# The Effects of True Triaxial Loading and Unloading Rates on the Damage Mechanical Properties of Sandstone

Man Wang [1], Weihang Du [2], Yingwei Wang [1], Xinjian Li [1,3], Liming Qiu [4], Beichen Yu [2], Zehua Niu [1,2] and Dongming Zhang [2,*]

[1] China Pingmei Shenma Group, State Key Laboratory of Coking Coal Exploitation and Comprehensive Utilization, Pingdingshan 467000, China
[2] State Key Laboratory of Coal Mine Disaster Dynamics and Control, Chongqing University, Chongqing 400000, China
[3] State Key Laboratory Cultivation Base for Gas Geology and Gas Control, Henan Polytechnic University, Jiaozuo 454000, China
[4] Key Laboratory of the Ministry of Education for Efficient Mining and Safety of Metal Mines, University of Science and Technology Beijing, Beijing 100083, China
[*] Correspondence: zhangdm@cqu.edu.cn

**Abstract:** Coal is the main energy source in China. In the process of coal resource mining, the surrounding rock of roadways is often in the complex stress environment of "three heights and one disturbance". At the same time, rocks in the stratum are often in a three-way unequal pressure state under the action of geological structure, and conventional rock mechanics tests cannot study the mechanical properties of rocks under actual stress conditions; thus, this is based on the self-developed true triaxial multifunctional fluid–structure coupling test system to study the damage mechanical Properties of Sandstone. The results are shown as follows: With an increase in loading rate, the peak damage $D_{cr}$ of sandstone decreases, but the initial damage $D_a$ increases in the elastic stage, and the brittleness of sandstone weakens. With the increase in the unloading rate, $D_{cr}$ increases, but $D_a$ decreases in the elastic stage, and the sandstone brittleness increases first, then decreases. In addition, the peak maximum principal strain $\varepsilon_{1max}$ first decreases rapidly and then slowly; the peak minimum principal strain $\varepsilon_{3max}$ increases first, then decreases slowly, and increases slowly; the peak intermediate principal strain $\varepsilon_{2max}$ decreases slowly; and the peak volume strain $\varepsilon_{vmax}$ increases rapidly first and then slowly with increases in the loading rate. With an increase in the unloading rate, $\varepsilon_{1max}$ increases rapidly first, then decreases slowly, then increases rapidly and finally increases slowly; $\varepsilon_{3max}$ first decreases slowly, then increases slowly, and finally decreases slowly; and $\varepsilon_{2max}$ increases slowly then decreases slowly. $\varepsilon_{vmax}$ decreases rapidly first and then increases slowly with increasing loading rate.

**Keywords:** true triaxial; loading and unloading rate; damage; sandstone brittleness

## 1. Introduction

Due to the surging exploitation of coal resources in recent decades, shallow energy cannot meet the growing demand of energy in China, which also makes the momentum of deep energy exploration and exploitation soar. At the same time, with the increasing depth of underground space engineering, safety controllability decreases, the stress conditions are more complex, energy mining is more difficult, and costs are higher [1].

With increasing mining depth under complex stress conditions, the roadway surrounding rock is more prone to instability failure in the process of coal excavation, resulting in casualties and economic losses. In addition, with increasing coal mining depth, it is easier to induce coal and gas outburst, rock burst, and other dynamic disasters. At present, the rock samples used in the tests cannot easily reflect the original occurrence environments and stress paths in deep mines; deep field engineering requires long periods of time, and

there is a lack of large-scale in situ monitoring research; therefore, the stress paths used in the laboratory tests are not consistent with the field [2]. In deep mines, the thick and hard bottom layer will produce higher mining stress, which will cause coal and rock roadways to produce larger deformation and even rock burst [3].

To better understand the mechanical properties of deep rocks and prevent or reduce the occurrence of these dangerous accidents, scholars at home and abroad have carried out a large number of studies in indoor laboratories, focusing on the mechanical properties of rocks under complex stress conditions. Xie et al. (2021) [4] conducted a conventional three-week test with different depths of ground stress and found significant differences in the physical and mechanical parameters of rocks at different depths. The results showed different brittleness characteristics of rocks with different occurrence depths. For sandstone with depth of 1600 m, the brittleness decreases with the increase in confining pressure on the whole, showing a transition from brittleness to ductility to strain hardening, and the post-peak plasticity gradually increases until it becomes completely plastic after the peak. Li et al. (2022a) [5] defined rock failure parameters by carrying out true triaxial tests and proposed a characterization method that could reflect the proportion of tensile and shear fractures in the rock failure process. Liu et al. (2021) [6] found that, compared with under the unloading path, granite required more energy when it was destroyed under the loading path in the true triaxial test, but it was more dangerous under the unloading path. Yin et al. (2019) [7] conducted a detailed study of the mechanical properties of sandstone under different loading and unloading rates using a true triaxial testing machine and found that tensile cracks were mostly concentrated on the unloading surface. Li et al. (2021a) [8] used a TRW-3000 true triaxial testing machine to carry out loading and unloading tests under different stress paths. Under DP criterion fitting, the cohesion and internal friction angle under loading conditions were higher than those under unloading conditions. Chu et al. (2022) [9] used MRI to analyze the pore and fracture expansion of coal samples after liquid nitrogen freezing and thawing. Quan et al. (2020) [10] used a true triaxial testing machine and a high-speed camera to study the mechanical properties of marble under different unloading rates and found that the failure process was more stable when unloading rate was lower, and the failure mode of marble changed from shear failure to shear tension failure with the reduction in minimum principal stress. Roohollah et al. (2020) [11] studied a wide range of rock properties and compiled a database, and they established a prediction model of rock burst maximum stress and risk index. Chu et al. (2019) [12], through triaxial cyclic loading and unloading tests on coal samples, found that the cumulative residual strain is related to the number of cycles. The more cycles, the greater the cumulative residual strain, but the relative residual strain gradually decreases, then stabilizes and finally rises sharply. Meanwhile, the total energy of a coal sample increases exponentially with increasing deviatoric stress. Zhai et al. (2020) [13] conducted the rock burst test under the condition of single-side airborne true triaxial loading and combined high-speed camera and SEM to study the results; they determined that the main reason for rock burst of different rock types is differences in the internal microscopic structures of rocks and their evolution under the different loading conditions. Danni et al. (2019) [14] found that the initial static stress is the main important factor in dynamic failure through true triaxial dynamic and static loading system research on rock dynamic failure. Su et al. (2016) [15] used a true triaxial rock burst testing machine to study rock samples at different high temperatures. The results showed that 300 °C was a critical point: The peak strength of rock samples changed little when the temperature was less than 300 °C, and the kinetic energy required by rock burst increased significantly when it was greater than 300 °C. Xiang et al. (2009) [16] studied the mechanical behavior of rock with a single structural plane under simulated excavation and support stress path using a true triaxial testing machine. The results showed that stress state, support strength and the parameters of the structural plane influenced the failure mode and support effect of rock with a structural plane under this stress path. Qiu et al. (2022) [17] used Hilbert-H and multiple analysis theory and studied the nonlinear characteristics of EMR and AE in re-coal cracking failure.

The results showed that the EMR and AE of coal cracking failure were related to the coal crack propagation process. Qiu et al. (2020) [18] established that the deformation and fracture of coal rock was caused by the accumulation of discrete fractures in rock samples. The time series obtained by the moving average method had a good correlation with the inner coal rock fractures and had obvious characteristics of coal instability and dynamic disaster precursor. Han et al. (2021) [19] carried out the true triaxial compression test of pre-cracked rock, and the results showed that the failure behavior of the compressive strength of the sample was related to the crack angle. The peak intensity decreased first and then increased with increasing crack priority angle. Li et al. (2019a) [20] used a true triaxial test system combined with CT scanning technology to study the mechanical properties of sandstone under different medium principal stress, and the results showed that the strength of sandstone increased first and then decreased with increases in medium principal stress. Dong et al. (2018) [21] used a true triaxial testing machine to study the mechanical properties of sandstone under biaxial compression and found that the fracture surface of sandstone specimen was parallel to the directions of intermediate principal stress and minimum principal stress, forming a large angle. Li et al. (2021b) [22] used a true triaxial rock burst testing machine to conduct surrounding rock failure tests on samples under pre-static load and dynamic disturbance and found that the threshold of rock burst occurrence and the frequency and failure degree of rock burst increased with the continuous increase of axial pressure. Li et al. (2022b) [23] used a true triaxial testing machine to study sandstone with holes and found that the holes significantly degraded the mechanical parameters of the specimen, and that the specimen entered the plastic yield stage in advance with decreasing peak strength. Fan et al. (2018) [24] studied the unloading failure strength of red sandstone under true triaxial conditions and found that the failure strength of red sandstone under rapid unloading condition decreased in different levels compared with that under loading condition. Wang et al. (2015) [25] studied the deformation and failure characteristics of fractured rock mass around the roadway under true triaxial conditions. The fracture angle had a great influence on the failure characteristics of rock mass; with the increase of the fracture angle, the compaction phenomenon was obvious, the dilatancy phenomenon showed a rising trend and the stress–brittle drop coefficient increased. Lee and Haimson (2011) [26] used a true triaxial testing machine to study granodiorite and found that rock strength increased with increases in intermediate principal stress. Wang et al. (2018) [27] performed conventional and true triaxial tests, and the results showed that in the first three stages of damage, evolution under the conditions of two different time and space distributions of acoustic emission activities was basically the same. Under the condition of CTT, the fracture surface of the test decreased with the increase of the confining pressure. However, under the condition of TTT, it first decreased and then increased with increasing intermediate principal stress. Wang et al. (2022) [28] used a true triaxial test system to study the mechanical properties of red sandstone under four different unloading stress paths and found that the octahedral shear stress was linearly correlated with the average effective stress. Hu et al. (2018) [29] used a true triaxial test system and DEM to study the characteristics and mechanism of rock burst induced by disturbed stress and found that under true triaxial test conditions, the test failure was mainly tensile splitting, and the generation of tensile cracks generally preceded the generation of shear cracks. In addition, DEM simulation results showed that weak dynamic rock burst was the result of tensile and shear failure. Zhao et al. (2021) [30] used a true triaxial test system to study the mechanical properties of sandstone under different loading and unloading rates and found that the bulk strain under a low unloading rate was mainly caused by axial compression and that rock damage was more serious at a high loading rate. Si et al. (2020) [31] conducted true triaxial testing on specimens with round holes of 50 mm diameter and found that the axial stress of hole wall failure increased with the increase of the loading rate, and the rock burst of the hole wall was more serious when the loading rate was lower. Zheng and Feng (2019) [32] used a true triaxial testing machine to study specimens with stress induction and found that with the decrease in the intermediate principal stress, Young's modulus decreased and the bilateral

deformation increased. Duan et al. (2017) [33] used DEM to study the failure mechanism of sandstone; the results show that macroscopic response to $\sigma_2$ played an important role. With the increase in $\sigma_2$, peak stress increased after the first drop, and damaged the angle increment obtained by numerical simulation increased. The influence of $\sigma_2$ on Young's modulus, however, increased as $\sigma_3$ and $\sigma_2$ had less of an effect on the mechanical properties. Ze et al. (2014) [34] used the true triaxial test system to study the effect of intermediate principal stress and found that the intermediate principal stress coefficient had a quadratic function relationship with rock strength. Li et al. (2019b) [35] used DEM to simulate the unloading process of materials with cracks. With the increase in the unloading rate, the more severe the failure was, the more the cracks split. Xiao et al. (2021) [36] carried out laboratory tests and discrete element simulation to study the mechanical properties of sandstone under the unloading condition of maximum principal stress. The results showed that: with the increase in maximum principal stress, the bearing limit of sandstone could be improved, but it was more likely to be destroyed when unloading. Kong et al. (2021) [37] constructed the dynamic constitutive equation of gas-bearing coal under impact load through an SHPB test, which clearly explained the influence of different conditions on the dynamic mechanical properties of coal samples.

Kong et al. (2019) [38] studied the damage evolution mechanism of gas-bearing coal in this process and the formation reasons of acoustic emission signals by carrying out loading tests on gas-bearing coal. Du et al. (2015) [39] revealed the influence of intermediate principal stress on the failure of platen by true triaxial unloading test. Zhao et al. (2014) [40] used the true triaxial strain explosion test system to study the strain explosion process by changing the unloading rate, and the results show that the strain explosion is more likely to occur when the unloading rate is high. Fan et al. (2020) [41] used the true triaxial test system to carry out the unloading test under cyclic load path. The research results show that: with the increase of cyclic load before unloading, the elastic modulus and unloading strength increase first and then decrease, and the failure mode of rock changes from tensile failure to mixed tensile shear failure. Lu et al. (2021) [42] through uniaxial compression, triaxial compression, and true triaxial unloading tests on basalt, the results show that the stress–strain curves under uniaxial and true triaxial compression show strain softening, and the stress–strain curves under triaxial compression show strain hardening. Miao et al. (2011) [43] conducted rock burst tests on granite with the true triaxial testing machine, and the results showed that the debris or irregular massive debris of granite was related to stress conditions and boundary conditions.

The results of the above research have a guiding significance for understanding coal dynamic disasters such as rock instability and rock burst in the process of coal mining, and provide an important reference for studying the mechanical properties under different actual triaxial loading and unloading conditions. In underground engineering construction and mining processes, the surrounding rock stress state is complex, to better meet the "three highs a disturbance" complex stress conditions; this research uses independent research and development of the multifunctional fluid–structure coupling triaxial test system for studying the sandstone under the different loading and unloading rates; the complex stress conditions of the instability of the roadway surrounding rock were examined in order to carry out the research.

## 2. Test Device and Scheme

### 2.1. Sample Device and Sample

Based on the self-developed true triaxial multifunctional fluid–structure coupling test system (as shown in Figure 1), mechanical tests of true triaxial sandstone under different loading and unloading rates are carried out. The system can provide the maximum pressure of 6, 6 and 4 MN in three directions to meet requirements of this test [44].

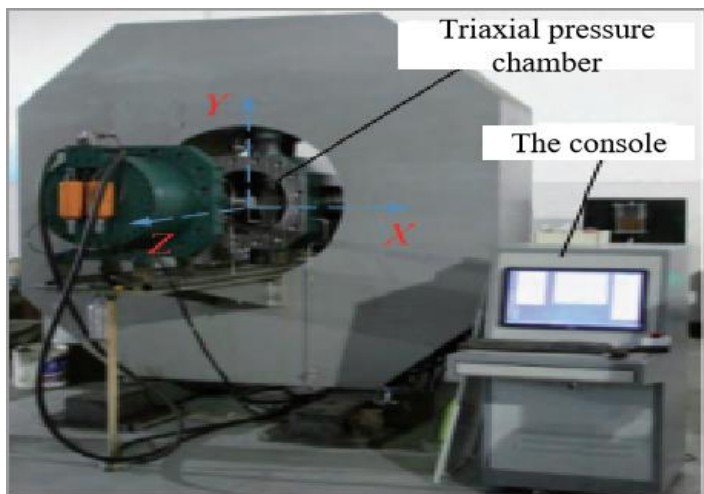

**Figure 1.** True triaxial fluid–structure coupling test system.

The samples taken in this test were from Mine No. 12 of PingMei ShenMa Groupin China. The sample is a $100 \times 100 \times 100$ mm cubic specimen with the end face flatness within 0.02 mm. Young's modulus E of the sandstone is 10.6 GPA, and Poisson's ratio v of the sandstone is 0.31. The apparent density is 2260 kg/m$^3$. This indicates that there are no obvious joints and fissures, in line with the standards of the International Society for Rock Mechanics.

*2.2. Test Scheme*

To better meet conditions of field stress, the loading and unloading methods of this test are $\sigma_3$ single-side unloading and $\sigma_1$ single-side loading. In this test, two sets of mechanical tests were performed on the sandstone at different loading and unloading rates, named group H and group G. For group H. The loading rate was kept the same and the unloading rate was changed. For group G, the unloading rate was kept the same and the loading rate was changed. The details of the test are shown in Table 1. The test stress path is shown in Figure 2, and proceeds as follows:

**Table 1.** Test schemes of different loading and unloading rates.

| Specimen Number | Loading Rate/(mm•$^{1-}$) | Unloading Rate/(kN•$^{s-}$) | Specimen Number | Loading Rate/(mm•$^{1-}$) | Unloading Rate/(kN•$^{s-}$) |
|---|---|---|---|---|---|
| H1 | | 0.2 | G1 | 0.001 | |
| H2 | | 1 | G2 | 0.003 | |
| H3 | 0.003 | 2.5 | G3 | 0.005 | 1 |
| H4 | | 3 | G4 | 0.008 | |
| H5 | | 5 | G5 | 0.012 | |

First, the stresses in the three directions are added synchronously at 40 MPa with a force control of 2 kN/s. The $\sigma_3$ remained unchanged, $\sigma_1$ and $\sigma_2$ continue to be simultaneously loaded to 60 MPa. The $\sigma_2$ and $\sigma_3$ remained unchanged and $\sigma_1$ continued to be loaded to 80 MPa. When the hydrostatic pressure was reached, the $\sigma_1$ was loaded in a displacement control way, and the $\sigma_3$ was unloaded at a single side in a force control way. When the sandstone sample broke, the test was stopped.

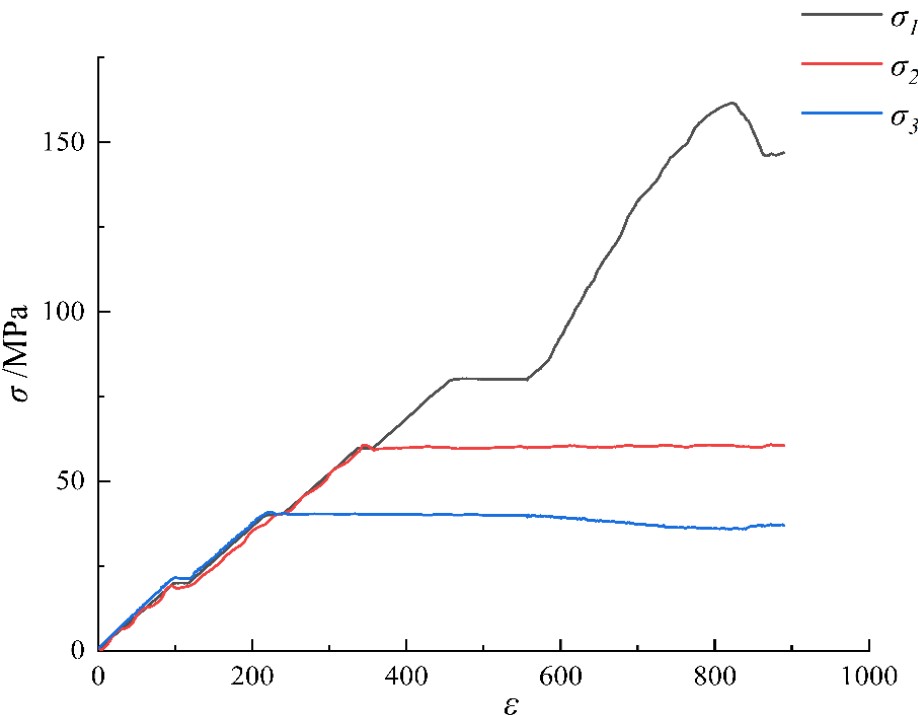

**Figure 2.** Stress–strain characteristics.

## 3. Experimental Results and Analysis

*3.1. Variation of Peak Strain and Peak Deviant Stress under True Triaxial Loading and Unloading Rates*

The peak stress and strain of rock are important indicators to measure the mechanical properties of rock, and important mechanical parameters can be obtained from them. In order to better analyze deformation characteristics of sandstone at different loading and unloading rates, the curves of peak strain-loading (unloading) rate (Figure 3a) and peak deviatoric stress-loading (unloading) rate (Figure 3b) were drawn.

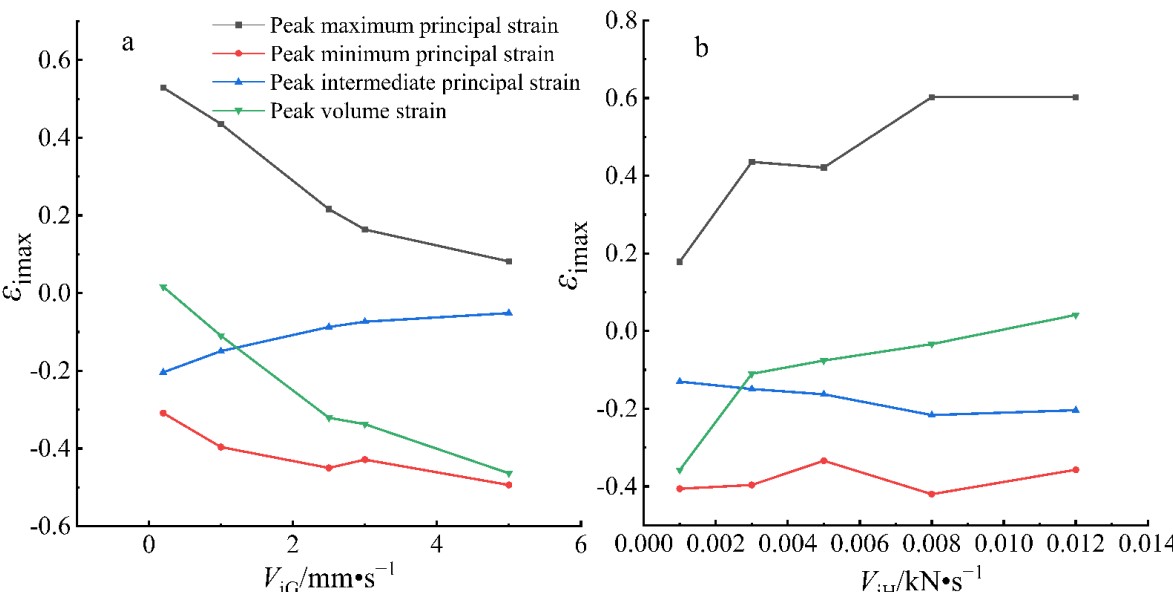

**Figure 3.** Peak strain-loading (unloading) rate curve: (**a**): different loading rate conditions, (**b**): different unloading rate conditions.

In Figure 3a, $\varepsilon_{\text{imax}}$ is the peak strain under different loading rates, and $Vi_G$ is the loading rate. It can be found from the figure that the peak maximum principal strain first decreases rapidly and then decreases slowly as the loading rate increases. With the increase in the loading rate, $\varepsilon_{3max}$ increases first and then slowly decreases, then slowly increases and then slowly decreases, and $\varepsilon_{2max}$ decreases slowly with the increasing loading rate, and $\varepsilon_{vmax}$ rapidly first and then slowly with the increase in the loading rate. Meanwhile, it can be found from Figure 3a that the variation of $\varepsilon_{vmax}$ is basically consistent with that of $\varepsilon_{1max}$. $Vi_H$ in Figure 3b is the unloading rate. It can be found from Figure 3b that $\varepsilon_{1max}$ increases rapidly first and then slowly decreases with the increase in the unloading rate, then increases rapidly and finally slowly increases. With the increase in the unloading rate, $\varepsilon_{3max}$ firstly decreases slowly, then increases slowly, and finally decreases slowly. $\varepsilon_{2max}$ increases slowly with the increase in the unloading rate. Eventually, it slowly decreased; $\varepsilon_{vmax}$ decreases rapidly first, and then slowly with the increase in the loading rate; the variation in $\varepsilon_{vmax}$ is similar to that of $\varepsilon_{1max}$.

It can be found from Figure 4a that the peak deviational stress first increases, then decreases rapidly, and finally slowly decreases as the loading rate increases. This is because the increase in loading rate accelerates the fracture rate of the sandstone specimen. Although the loading rate is very high at this time, the time required for the rock to fracture is also reduced, and the increase in the peak deviational stress is greatly reduced when the unloading rate is high. It can be found from Figure 4b that the peak deviationic stress first increases rapidly, then slowly, then rapidly, and finally slowly decreases with the increase in the unloading rate. When the unloading rate is low, the increase in peak deviationic stress is higher.

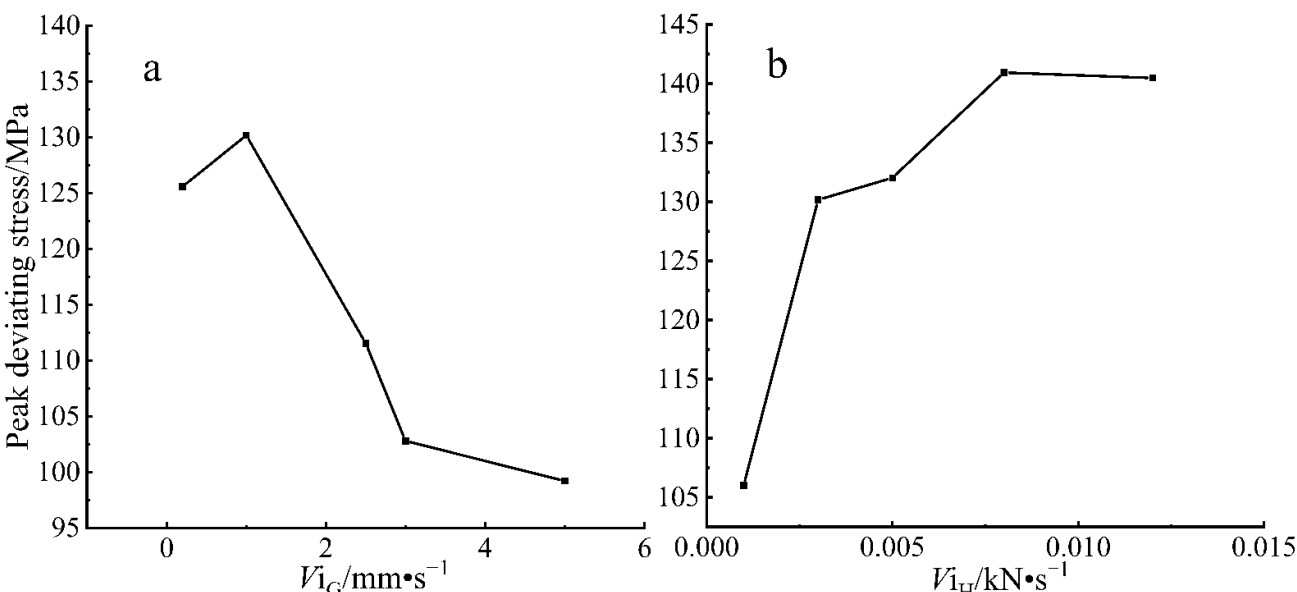

**Figure 4.** Peak deviatoric stress-loading (unloading) rate curve: (**a**): different loading rate conditions, (**b**): different unloading rate conditions.

To better understand the influence of the increasing loading and unloading rate on peak strain and deviational stress, the slope $K$ of the line between the other four points is plotted against G1 and H1, respectively. The value of $K$, based on G1 and H1, shows the effect of the increasing loading and unloading rate on peak strain and deviational stress. At the same time, $K$ (displacement generated at unit loading and unloading rate) is an indicator to analyze the influence of increasing loading and unloading rate on rock deformation and strength. Table 2 shows the values of $K$ for different loading rates, and a negative value of $K$ indicating a decreasing displacement. $K_{\varepsilon1}$ represents the response of the increasing loading rate to the displacement in the $\sigma_1$ direction. $K_{\varepsilon2}$ represents the response of the increasing loading rate to the displacement in the $\sigma_2$ direction. $K_{\varepsilon3}$ represents the response

of the increasing loading rate to the displacement in the $\sigma_3$ direction. $K_{\varepsilon v}$ represents the response to peak volume strain after increasing the loading rate. $K_\sigma$ represents the response to deviatoric stress as the loading rate increases.

**Table 2.** Slope $K$ of the line between the other four points and the base point under the condition of the loading rate.

| Slope Number | $K_{\varepsilon 1}$ | $K_{\varepsilon 2}$ | $K_{\varepsilon 3}$ | $K_{\varepsilon V}$ | $K_\sigma$ |
|---|---|---|---|---|---|
| G2-G1 | −0.1168 | 0.0683 | −0.1096 | −0.1576 | 5.7574 |
| G3-G1 | −0.1360 | 0.0509 | −0.0614 | −0.1466 | −6.1034 |
| G4-G1 | −0.1304 | 0.0466 | −0.0428 | −0.1263 | −8.1351 |
| G5-G1 | −0.0932 | 0.0318 | −0.0386 | −0.1000 | −4.5491 |

It can be found from Table 2 that $K_{\varepsilon 1}$ decreases first and then increases with the increase in the loading rate, indicating that the response to $\varepsilon_1$ becomes faster and then slowly decreases with the increase in the loading rate, and the response to $\varepsilon_1$ becomes slower with the increase in the loading rate. With the increase in the loading rate, $K_{\varepsilon 2}$ decreases and $K_{\varepsilon 3}$ increases gradually. It shows that the response of $\varepsilon_2$ to the increase in the loading rate is slower and the response of $\varepsilon_3$ to the increase in the loading rate is slower. With the increase in the loading rate, $K_{\varepsilon V}$ increases gradually, which means that $\varepsilon_v$ responds more and more slowly to the increase in the loading rate. In addition, it can be found from the table that the absolute values of $K_{\varepsilon 1}$, $K_{\varepsilon 2}$ and $K_{\varepsilon 3}$ are the largest; thus, the fracture of the sandstone specimen is dominated by the increase in $\varepsilon_1$. It can be found from Table 3 that $K_{\varepsilon 1}$ and $K_{\varepsilon V}$ decrease with the increase in the unloading rate. $\varepsilon_v$ and $\varepsilon_1$ respond slowly to the increase in the unloading rate. Although $K_{\varepsilon 2}$ and $K_{\varepsilon 3}$ show no obvious change rule, the change degree is not significant. At the same time, it can be found from the table that the absolute values of $K_{\varepsilon 1}$, $K_{\varepsilon 2}$ and $K_{\varepsilon 3}$ are the largest, which means that under different unloading rates, the increase in $\varepsilon_1$ also leads to the failure of sandstone specimens.

**Table 3.** Slope $K$ of the line between the other four points and the base point under the condition of unloading rate.

| Slope Number | $K_{\varepsilon 1}$ | $K_{\varepsilon 2}$ | $K_{\varepsilon 3}$ | $K_{\varepsilon V}$ | $K_\sigma$ |
|---|---|---|---|---|---|
| H2-H1 | 128.57 | 9.57 | 4.88 | 123.885 | 11,970.26 |
| H3-H1 | 60.61 | 8.1425 | 18.025 | 70.49 | 6503.73 |
| H4-H1 | 60.59 | 12.3028 | −1.9843 | 46.2986 | 4988.28 |
| H5-H1 | 42.41 | 6.6945 | 4.4455 | 36.3055 | 3132.19 |

*3.2. Damage Characteristics of Sandstone under True Triaxial Loading and Unloading Rates*

In underground space engineering such as tunnels and roadways, damage to sandstone is associated with the conditions of excavation. To explore the damage properties of sandstone under different opening conditions, real triaxial tests were performed at different loading and unloading rates. Under realistic triaxial loading and unloading rates, the extent of damage to the sandstone increases with the increase in time under three-dimensional stress. For the intact sandstone, the initial damage variable $D = 0$. However, the sandstone specimens we took in the laboratory were both macroscopically and microscopically defective. To obtain more accurate damage variables, [45] Qin et al. (2018) optimized the damage state of the traditional constitutive model which was based on Weibull statistical damage mechanics. The damage value and the initial damage coefficient $k$ were obtained for each characteristic point:

$$m = \frac{1}{\ln^E - \ln^{E_M}} \tag{1}$$

$$\sigma_{\max} = E\varepsilon_{\max}\left(ke^{-\frac{1}{m}}\right) \tag{2}$$

According to Equation (2), the initial damage coefficient $k$ can be obtained:

$$k = \frac{\sigma_{\max}}{E\varepsilon_{\max}} e^{\ln E - \ln E_M} \tag{3}$$

where $m$ is the mean value of the materials; $E$ is the elastic modulus of rock materials; $E_M$ is the secant modulus of the peak, $E_M = \frac{\sigma_{\max}}{\varepsilon_{\max}}$; $\varepsilon_{\max}$ is the maximum principal strain of the sandstone peak. $\sigma_{\max}$ is the peak maximum principal stress. Table 4 shows the material coefficient m and the damage coefficient k for the sandstone specimens selected for this test.

**Table 4.** Material coefficient and damage coefficient of the sandstone specimen.

| Specimen Number | Damage Coefficient | Degree of Material Mean | Specimen Number | Damage Coefficient | Degree of Material Mean |
| --- | --- | --- | --- | --- | --- |
| G1 | 1.0000 | 2.0000 | H1 | 0.9999 | 3.0894 |
| G2 | 0.9858 | 2.2216 | H2 | 0.9999 | 2.2216 |
| G3 | 1.0000 | 2.3100 | H3 | 0.9999 | 2.1465 |
| G4 | 0.9999 | 2.8277 | H4 | 0.9999 | 1.7948 |
| G5 | 0.8892 | 3.0415 | H5 | 0.9999 | 1.6428 |

Damage values for each characteristic point of the sandstone specimens can be obtained from the damage coefficient, material mean value, peak maximum principal stress and peak maximum principal strain:

$$D_{cr} = 1 - ke^{\frac{-1}{m}}\varepsilon \tag{4}$$

$$D_{th} = 1 - ke^{\frac{-1}{m}(1-D_{cr})^m} \tag{5}$$

$$D_c = 1 - ke^{\frac{-1}{m}(1-D_{cr})^{2m}} \tag{6}$$

$$D_b = 1 - ke^{-\left(\frac{1}{m}\right)^m \frac{1}{m}(1-D_{cr})^m} \tag{7}$$

$$D_a = 1 - e^{(1-D_{cr})^m \frac{-D_{cr}^m}{m}} \tag{8}$$

where $D_{cr}$ is the peak damage value; $D_{th}$ is the elastic limit point damage value, the end point of the elastic damage; $D_b$ is the damage value at the half peak intensity point; $D_a$ is the initial damage value for linear elasticity. According to Formulas (4)–(8) above, the damage values of each feature point under different loading and unloading rates are calculated as is shown in Tables 5 and 6. Figure 5 shows the distribution of each feature point.

**Table 5.** Damage values of feature points under different loading rates.

| Specimen Number | $D_{cr}$ | $D_{th}$ | $D_c$ | $D_b$ | $D_a$ |
| --- | --- | --- | --- | --- | --- |
| G1 | 0.8636 | 0.00069 | 0.0000015 | 0.000021 | 0.00044 |
| G2 | 0.7223 | 0.0258 | 0.0015 | 0.0044 | 0.0126 |
| G3 | 0.7358 | 0.0264 | 0.0015 | 0.0052 | 0.0137 |
| G4 | 0.6548 | 0.0793 | 0.0122 | 0.0284 | 0.0379 |
| G5 | 0.6440 | 0.1055 | 0.0330 | 0.0481 | 0.0526 |

**Table 6.** Damage values of characteristic points under different unloading rates.

| Specimen Number | $D_{cr}$ | $D_{th}$ | $D_c$ | $D_b$ | $D_a$ |
| --- | --- | --- | --- | --- | --- |
| H1 | 0.6791 | 0.0502 | 0.0053 | 0.0128 | 0.0235 |
| H2 | 0.7223 | 0.0258 | 0.0015 | 0.0044 | 0.0126 |
| H3 | 0.8581 | 0.0038 | 0.00005 | 0.00069 | 0.0033 |
| H4 | 0.8918 | 0.0007 | 0.000002 | 0.000035 | 0.00048 |
| H5 | 0.9477 | 0.00005 | 0.0000001 | 0.000002 | 0.00004 |

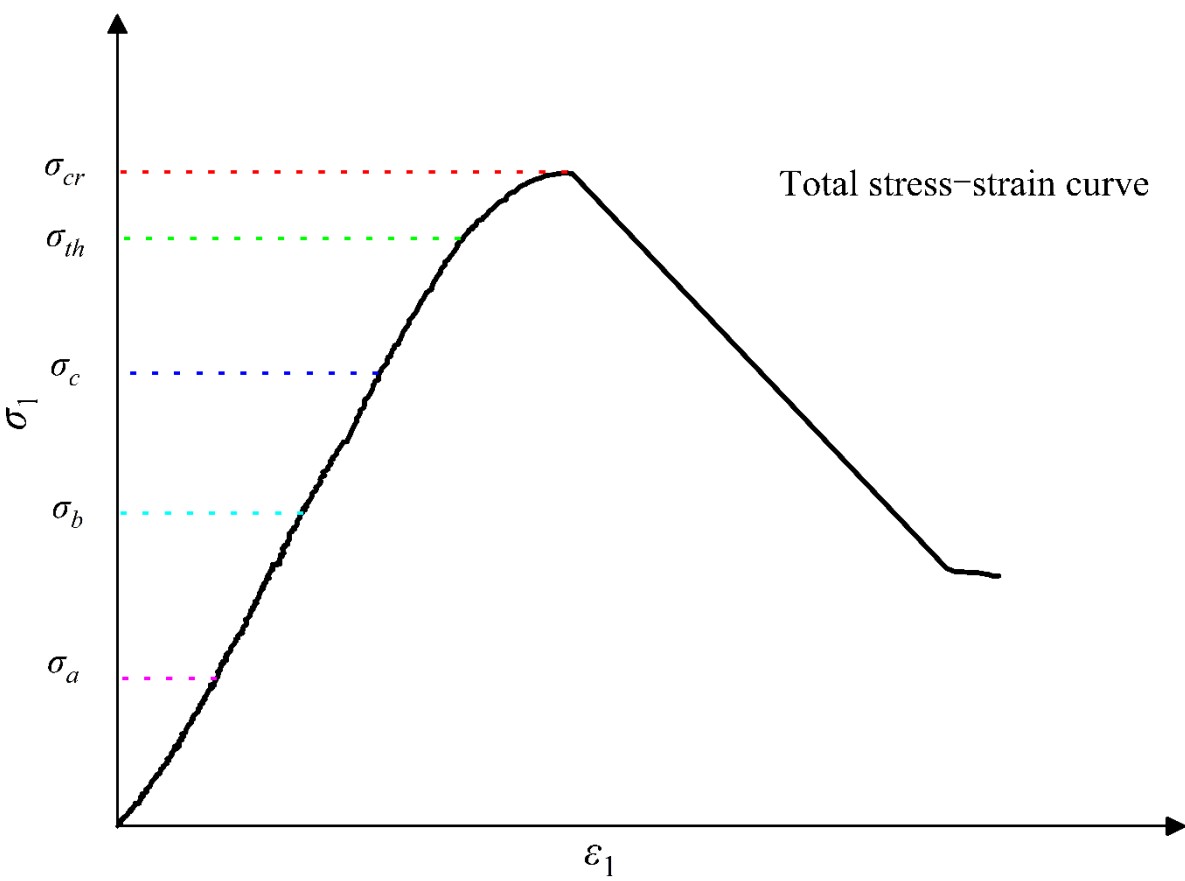

**Figure 5.** Distribution diagram of each feature point.

As shown in Table 5, under the conditions of different real triaxial loading rates, such as the direction of $\sigma_1$ loading and the direction of $\sigma_3$ unloading, the sandstone specimen has little damage, with no microcrack occurring before the $D_c$ line elastic damage ending point. The rate of native crack propagation is slow. As the $\sigma_1$ direction continues loading and the $\sigma_3$ direction continues unloading, the damage value increases significantly at the elastic limit point of $D_{th}$, but the damage value of sandstone specimen H1 remains small and can be ignored at this point; thus, crack growth and the development of H1 occur after the elastic limit point of $D_{th}$. The remaining specimens are considered to have begun to expand before $D_{th}$; thus, it can be found that the loading rate will affect the propagation rate of the fissures in the rock. However, as can be seen from Figure 6 (*D* is damage value; V loading rate) that shows the loading rate—damage value curve of each characteristic point, the damage value of peak point decreases with the increase in the loading rate; G1 with the lowest loading rate has the largest peak damage value. This is probably because the low loading rate is more conducive to the increase in the number of cracks. At the same time, it can be found from Table 5 that increasing the loading rate when it is low has a larger effect on the peak damage value of sandstone than increasing the loading rate when it is high. In addition, $D_{th}$, $D_c$, $D_b$ and $D_a$ all increase with the increase in the loading rate. When the loading rate is low, $D_{th}$, $D_c$, $D_b$ and $D_a$ are small. This does not favor the initial damage of the internal structure of sandstone during the elastic stage. Since damage to the rock begins to expand from internal cracks, the macro damage to the sandstone specimen is easier and faster when the unloading rate is higher. It can be seen from Table 5 that the damage of sandstone specimens at different loading rates is mainly in the stage from $D_{th}$ to $D_{cr}$.

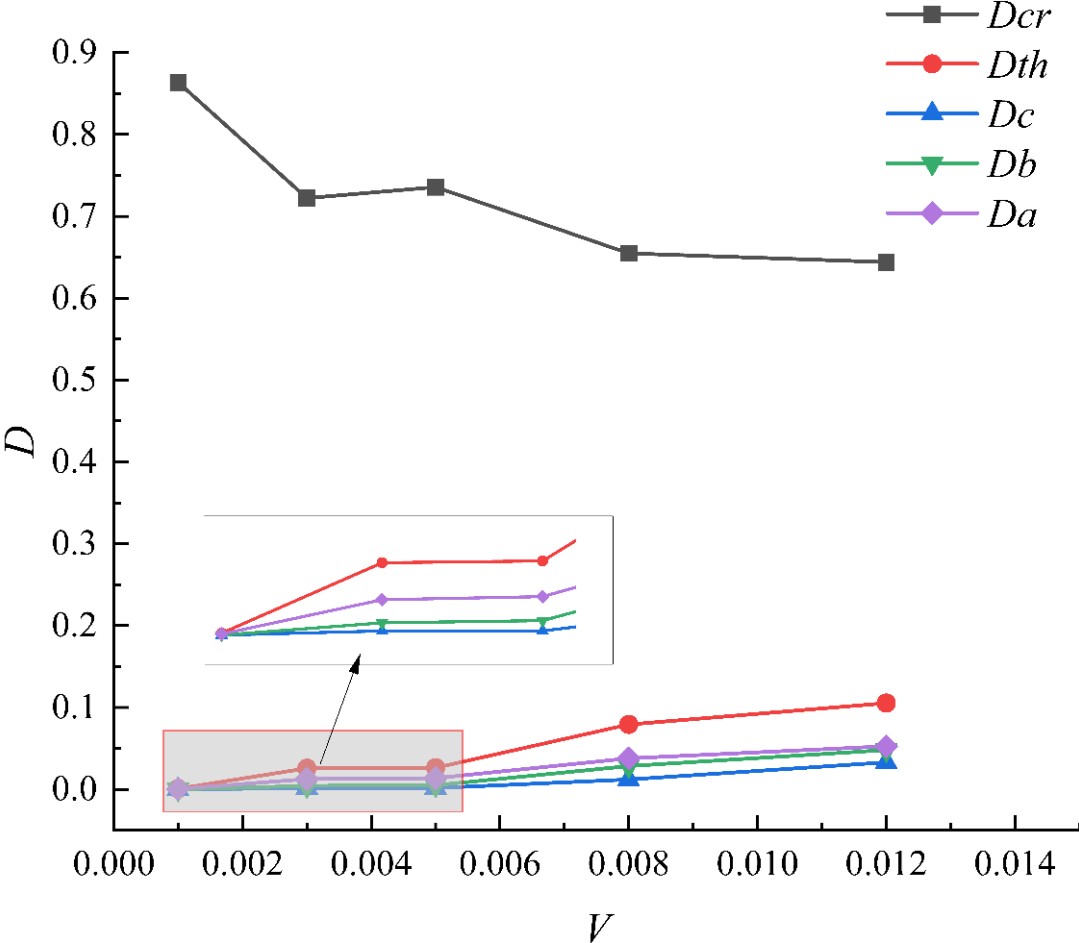

**Figure 6.** Loading rate–damage value curve of each characteristic point.

It can be found from Table 6, under the conditions of different loading and unloading rates in the true triaxial, with the loading in the $\sigma_1$ direction and unloading in the $\sigma_3$ direction, that sandstone basically has no damage at the elastic limit point. As the direction of $\sigma_3$ continues unloading and the direction of $\sigma_1$ continues loading, the only H1 and H2 with a low unloading rate occur an obvious increase before the $D_{th}$ elastic limit point. Meanwhile, from Figure 7 (D is damage value; V1 is the unloading rate) the unloading rate can be seen—the damage value curve of each characteristic point shows that $D_{th}$, $D_c$, $D_b$ and $D_a$ all decrease with the increase in the unloading rate. With the increase in the unloading rate, the initial damage rate of the internal structure of sandstone is slower, and the reduction in the unloading rate causes more damage to the sandstone specimen during the initial elastic stage. However, it can be seen from the figure that the peak damage value increases with the increase in the unloading rate; thus, the internal damage of sandstone can be accelerated when the unloading rate is low, but the number of cracks in macroscopic failure will decrease. As the failure of rock begins to expand from the internal cracks, the macro failure of the sandstone specimen is easier and faster when the loading rate is low, but the damage to sandstone will be reduced. It can be seen from Table 6 that the damage of the sandstone specimens with different unloading rates is mainly in the stage from $D_{th}$ to $D_{cr}$.

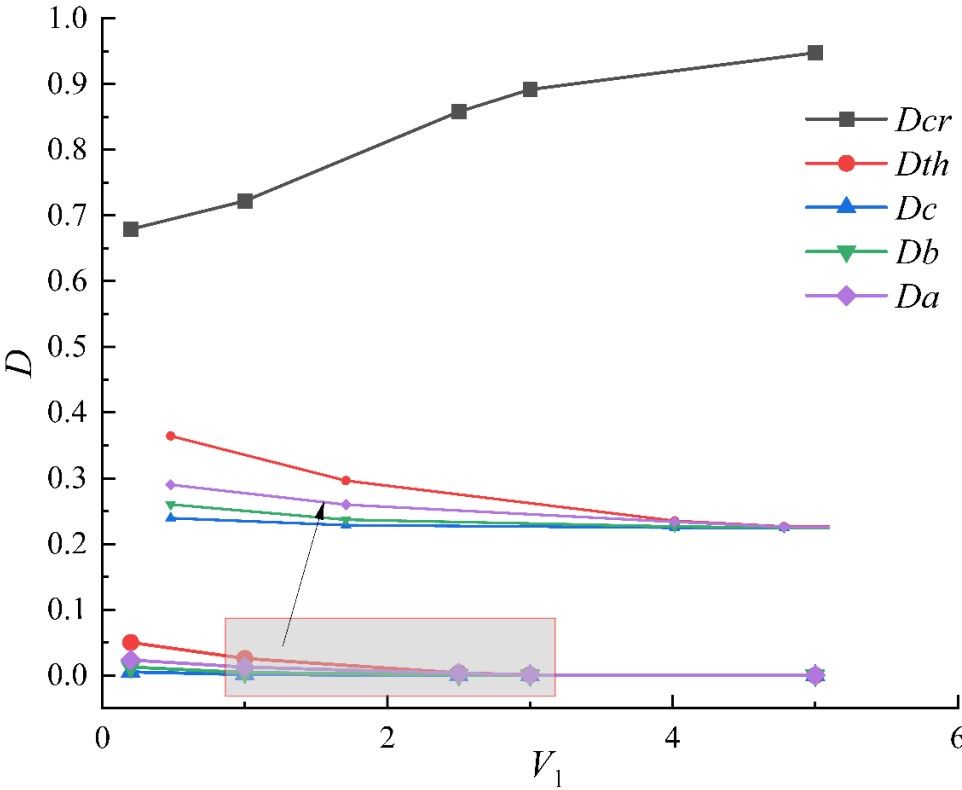

**Figure 7.** Damage curves of unloading rate and each characteristic point.

Deviational stress is the main factor responsible for the macroscopic failure of the internal structure. To analyze the damage response capacity of deviational stress to rock under different loading and unloading rates, a deviational stress damage compliance $\Delta q$ is introduced:

$$\Delta q = \frac{D_{cr}}{\sigma_{1-3}} \tag{9}$$

where $\sigma_{1-3}$ is the deviatoric stress difference between loading and unloading starting point and peak point; $D_{cr}$ is peak damage value, according to Equation (9). The deviational stress calculated is shown in Figures 8 and 9.

It can be found from Figure 8 that as the loading rate increases, the deviational stress damage compliance first decreases and finally increases, and the deviational stress response capacity becomes slower. However, when the loading rate decreases to 0.12 mm$\cdot^{1-}$, the response capacity of deviational stress to sandstone increases again, because when the loading rate is too high, the bearing capacity of sandstone decreases.

From Figure 9, it is shown that with the increase of unloading rate of deviator stress damage increase compliance, and the increasing rate is stable, but when the unloading rate is lower, increased compliance deviatoric stress injury after unloading rate change is small; therefore, when the unloading rate increased to a certain value, the deviatoric stress damage compliance of sandstone began to stably increase and the response ability to sandstone damage stably increased.

### 3.3. Brittleness Characteristics of Rock under True Triaxial Loading and Unloading Rates

#### 3.3.1. Stress Brittle Drop Factor

In traditional materials, materials that produce large deformation but do not crack are ductile materials with good ductility, or, on the contrary, are brittle materials. For rock materials, the key to distinguishing brittle from ductile is the type of rock failure; that is, the form of the rock failure process itself. Nowadays, there are many studies on rock brittleness

under unicycle compression or triaxial confining pressure, but there are few studies on rock brittleness under true triaxial condition [46].

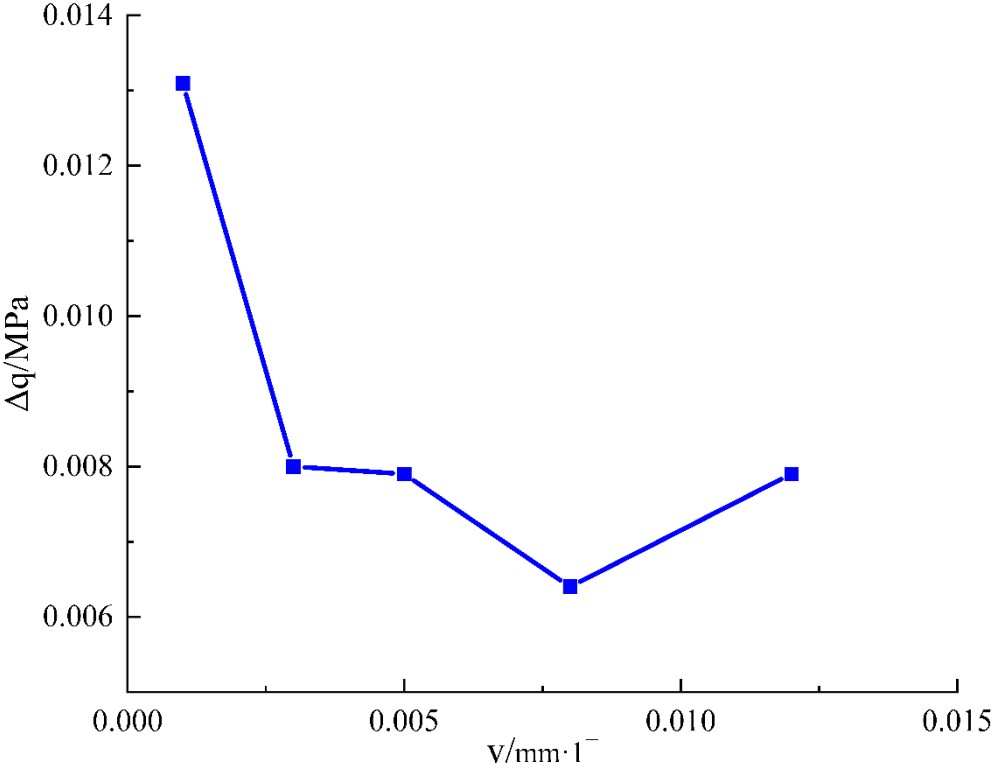

**Figure 8.** Deviatoric stress damage flexibility–loading rate curve.

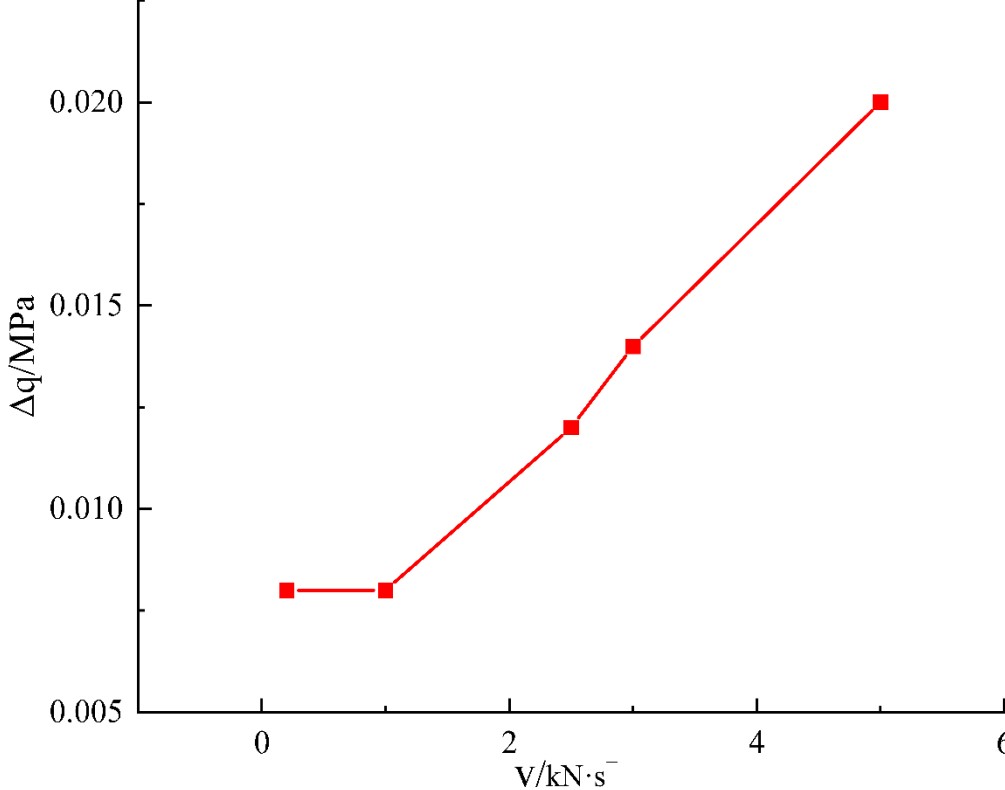

**Figure 9.** Deviatoric stress damage compliance–unloading rate curve.

Figure 10 shows the stress–strain characteristic curve of specimen H3, calculated to be 47.12%. Based on the stress-brittle drop process of the non-vertical drop model derived from Ge (1997), Shi et al. (2006) [47,48] determined the stress-brittle drop coefficient *R* in combination with the typical stress–strain curve generalization diagram of rock (Figure 10):

$$R = \frac{b}{a} \tag{10}$$

where *a* and *b* are strain-related parameters, $a = \varepsilon_p - \varepsilon_n$, $b = \varepsilon_b - \varepsilon_p$, $\varepsilon_p$ is the maximum principal strain of peak strength, $\varepsilon_b$ is the maximum principal strain of residual strength, and $\varepsilon_n$ is the strain of residual strength corresponding to the initial loading stage. According to Equation (10), the smaller *R* is, the more serious the stress brittle failure of rock is.

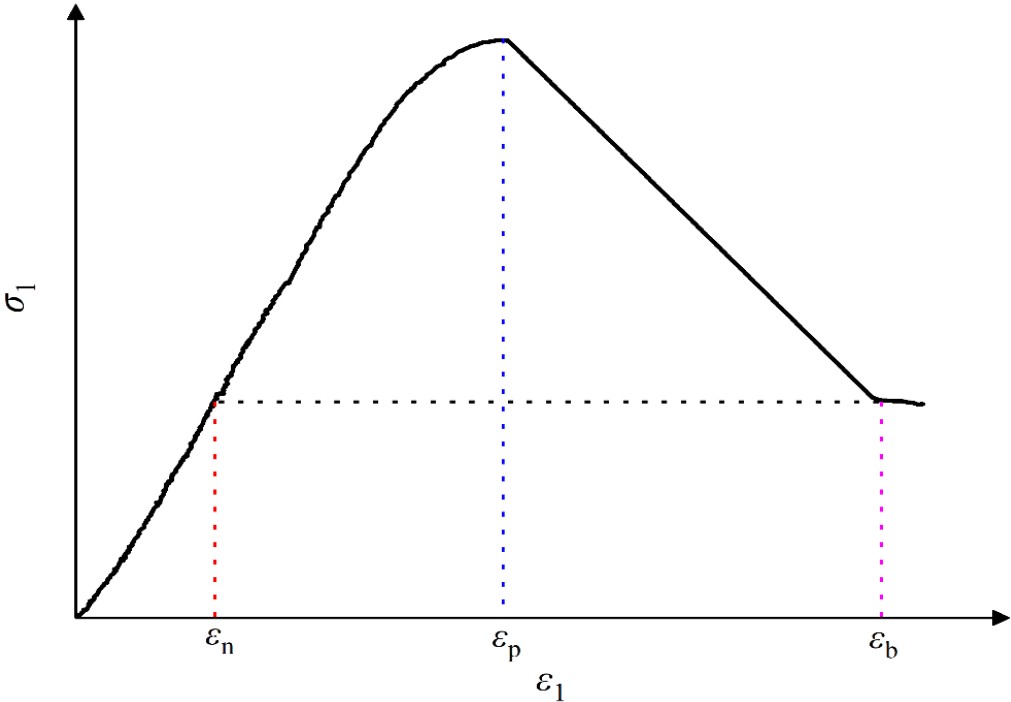

**Figure 10.** Classical stress–strain curves of brittle rocks.

3.3.2. Variation Characteristics of Specimen Characteristic Parameters under Different Loading Rates

Under the same initial stress path and different loading and unloading rates, the stress brittle drop coefficients and are function of the characteristic paramecium and the loading rates, which can be expressed as follows:

$$R = \frac{\varepsilon_b(v) - \varepsilon_p(v)}{\varepsilon_p(v) - \varepsilon_n(v)} \tag{11}$$

In the experiment, the characteristic parameters at different loading rates are shown in Figure 11. As can be seen from Figure 11, the displacement of the maximum principal strain direction at the peak strength point, the displacement of the residual strength point and the residual strength point all showed an increasing trend for the displacement of the loading section, and the increasing trend was basically the same. The dotted line in the figure fitted curve: $\varepsilon_n = -3714.729v^2 + 0.254 + 91.639v$; $\varepsilon_p = -4594.288v^2 + 0.519 + 94.123v$; $\varepsilon_b = -5046.028v^2 + 0.847 + 97.952v$.

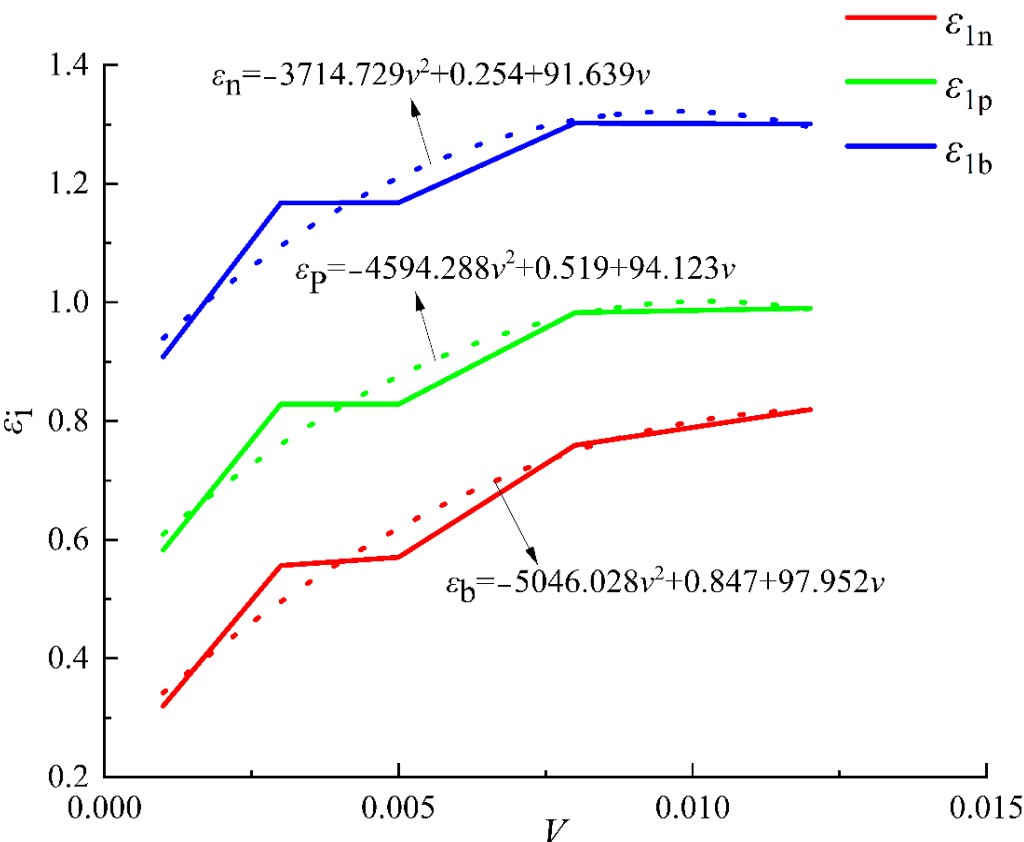

**Figure 11.** Curves of the relationship between residual strength point, peak point and residual strength point corresponding to the maximum principal strain displacement of loading segment point and loading rate under different loading rates.

Combined results:

$$R(v) = \frac{-451.74v^2 + 0.328 + 3.829v}{-879.559v^2 + 0.265 + 2.484v} \tag{12}$$

Equation (11) is the relation between stress brittle drop coefficient and loading rate at different loading rates. Through Equation (11), the stress brittle drop coefficient at different loading rates can be obtained, as is shown in Figure 12. With the increase in the loading rate, the stress brittle drop coefficient increases, and sandstone brittleness weakens.

### 3.3.3. Variation Characteristics of Specimen Characteristic Parameters under Different Unloading Rates

Under the same initial stress path and different unloading rates, the stress–brittle drop coefficients and are functions of the characteristic parameters of the specimen, which can be expressed as follows:

$$R = \frac{\varepsilon_b(v) - \varepsilon_p(v)}{\varepsilon_p(v) - \varepsilon_n(v)} \tag{13}$$

In the experiment, the characteristic parameters at different unloading rates are shown in Figure 13. As can be seen from Figure 13, the displacement of the maximum principal strain direction at the peak strength point, the displacement of the residual strength point and the residual strength point all show an increasing trend for the displacement of the loading section, and the increasing trend was basically the same. The dotted line in the figure fitted curve: $\varepsilon_n = 0.02046v^2 + 0.96941 - 0.18168v$; $\varepsilon_p = 0.01959v^2 + 1.21853 - 0.20361v$; $\varepsilon_b = 0.02705v^2 + 1.55467 - 0.25897v$.

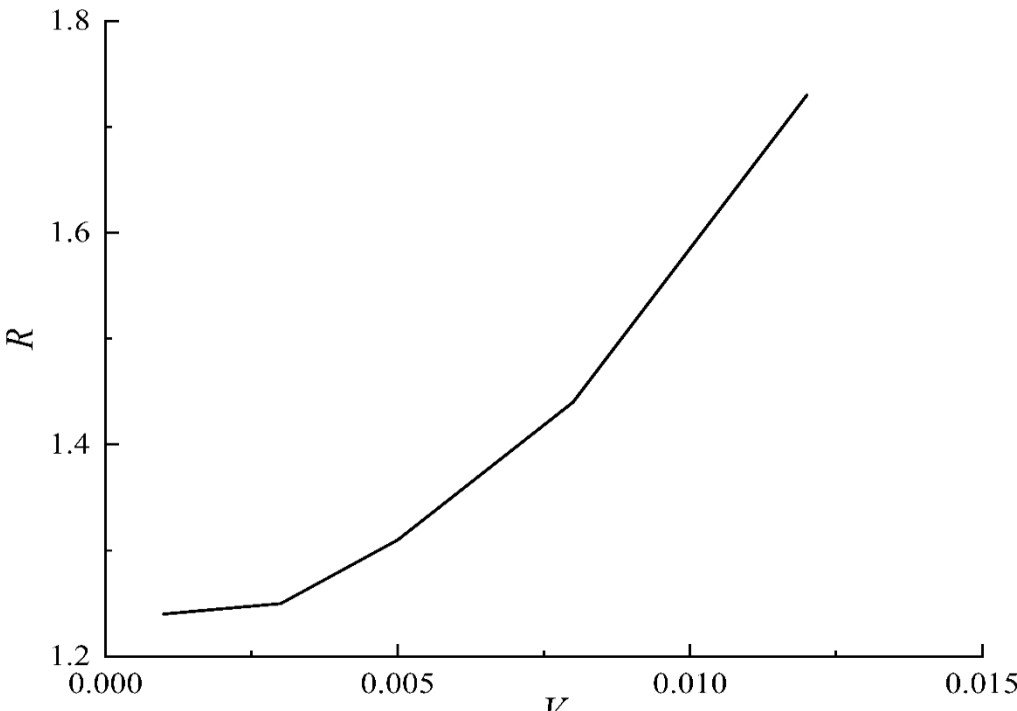

**Figure 12.** The relation curve between stress brittle drop coefficient and loading rate at loading rate.

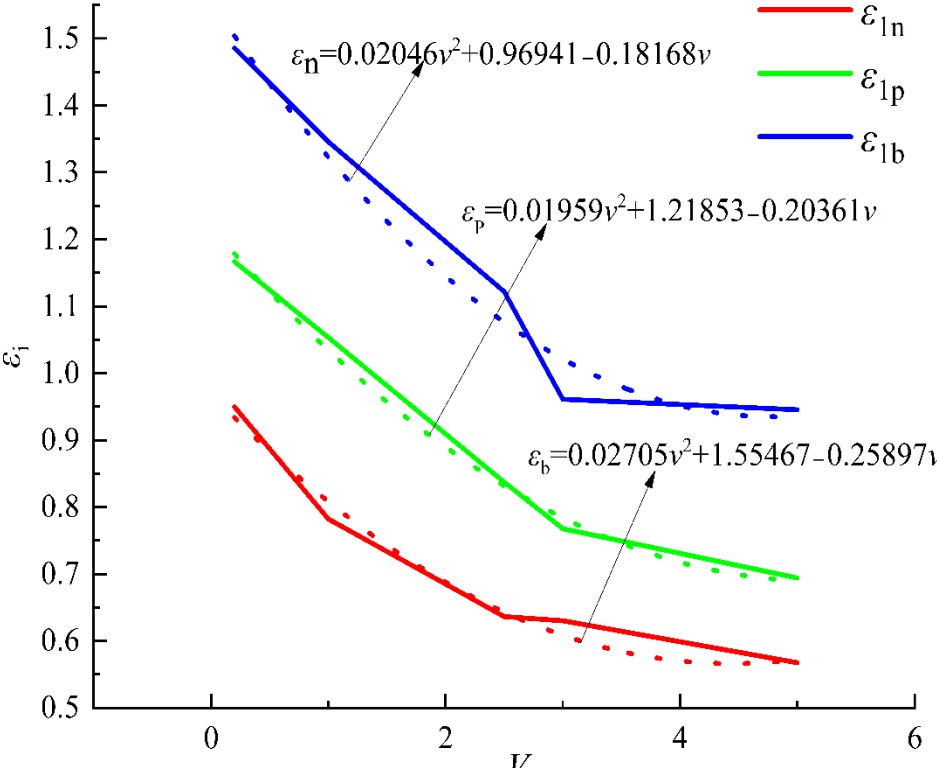

**Figure 13.** Curves of the relationship between the maximum principal strain displacement and unloading rate of the loading segment corresponding to residual strength point, peak point and residual strength under different unloading rates.

Combined results:

$$R(v) = \frac{0.00746v^2 + 0.33614 - 0.05536}{-0.00087v^2 + 0.24912 - 0.02193v} \tag{14}$$

Equation (14) is the relationship between stress brittle drop coefficient and unloading rate under different unloading rates. According to Equation (14), the stress brittle drop coefficient under different unloading rates can be obtained. As is shown in Figure 14, with the increase in the unloading rates, the stress brittle drop coefficient first decreases and the sandstone brittleness increases, and then the sandstone brittleness decreases. When the unloading rate is low, its brittleness remains relatively stable, and when the unloading rate reaches 5 kN/s, it decreases sharply, which is caused by the vertical drop in the stress–strain curve after the peak, which causes the abnormal increase in $\varepsilon_b - \varepsilon_P$. Therefore, this method is not suitable for rock brittle assessment at high unloading rates.

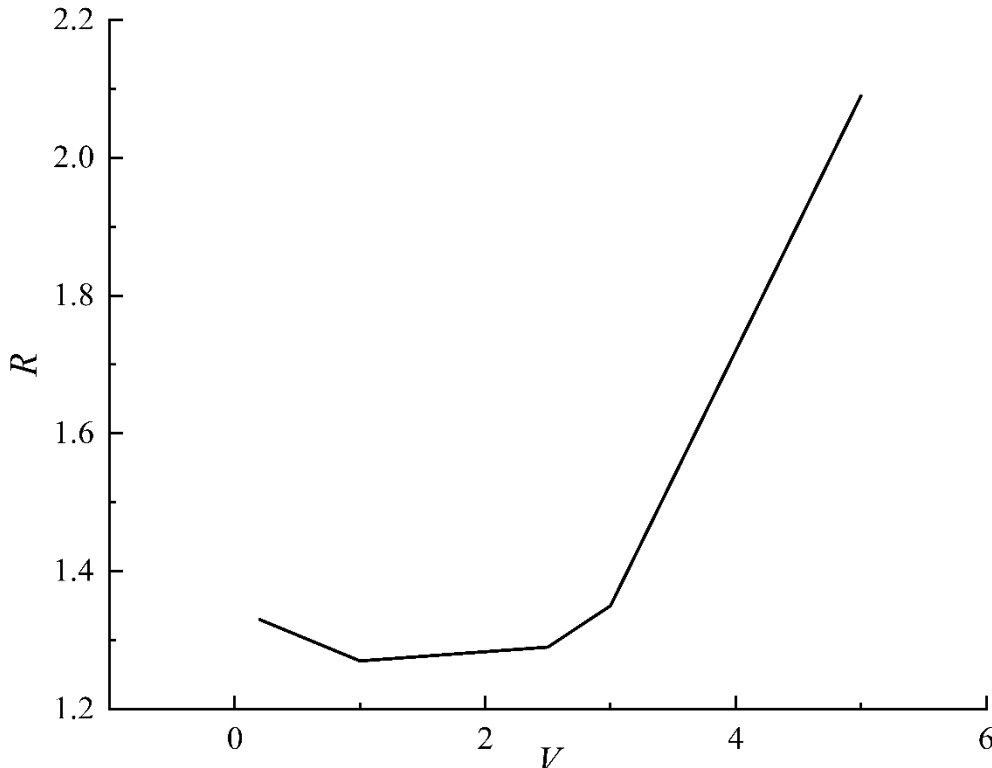

**Figure 14.** The relation curve between stress brittle drop coefficient and unloading rate at different unloading rate.

## 4. Discussion

In underground engineering construction, the risk of rock mass instability is often reduced by slowing down the excavation rate and reducing the excavation footage, the essence of which is to adjust the rate of surrounding rock stress loading and unloading caused by excavation, so as to reduce the possibility of rock mass instability and rock burst [49]. During underground excavations, high-stress areas are more prone to failure, and rock mass instabilities and failures are frequent. Under the condition of high-stress single-side unloading, the surrounding rock failure is the composite failure of tensile, splitting and shear [50]. Huang and Huang, 2010, [51] conducted a triaxial unloading test in the laboratory and found that the unloading rate and initial confining pressure have a great influence on the brittleness and tensile fracture characteristics of rock, and this influence is more obvious when unloading at high speed and high initial confining pressure. Whether the roadway is stable is closely related to the supporting conditions and methods. Under the condition of true triaxial single-plane in-flight test, the failure mode of rock samples

changes from brittle cracking to dynamic rock burst failure with the increase in the failure rate of support [52].

During coal mining and roadway excavation, the surrounding rock mass subjected to disturbance stress undergoes instability failure, which is different for different degrees of excavation depth and tunneling rate. According to the analysis in Section 3.2, the peak damage value of sandstone decreases with the increase in loading rate for different loading rates. In this case, the likelihood of inducing rock mass instability is lower, and the excavation strength and turning rate can be appropriately increased to ensure rock mass stability during mine excavation and roadway excavation. The damage value of the sandstone increases with the off-loading rate in the case of a two-fold differential off-loading rate. At this point, the likelihood of inducing rock mass instability increases. We can reduce the mining intensity and reduce the running footage of the roadways through the coal to ensure the stability of the rock mass. In addition, when the perturbation stress of the surrounding rock mass is too high, the accumulated strain energy of the empty rock mass is suddenly and violently released, resulting in an explosion-like brittle fracture of the rock mass. Rockfalls can cause large amounts of rock to fall and produce loud sounds and gas waves that can not only destroy mines but also endanger buildings on the surface. According to the analysis in Section 3.3, the brittle failure of sandstone weakens with increasing loading rate for different loading rates. However, the brittle failure of the sandstone is enhanced as the loading rate is increased at different offloading rates. As mentioned above, the offloading rate can be reduced at different offloading rates to ensure the stability of the rock mass. However, according to the analysis in Section 3.3, the brittle failure of sandstone is stronger at low offloading rates. Therefore, in the actual process of mine production and road tunneling, the appropriate excitation intensity and turning rate must be chosen to achieve safe and efficient production.

## 5. Conclusions

To ensure the safety of underground space excavation and to provide a theoretical basis for its use in laboratory tests, mechanical tests of sandstone real triaxial at different loading and unloading rates were performed based on the self-developed multi-function real triaxial test system. The results are shown as follows:

(1) With the increase in the loading rate, the peak $\varepsilon_1$, $\varepsilon_3$ and $\varepsilon_v$ decrease, and the peak $\varepsilon_2$ increases, and the peak deviational stress increases first and then decreases. With the increase in the unloading rate, the peak $\varepsilon_1$ and $\varepsilon_v$ increase, and the peak $\varepsilon_2$ decreases first and then increases, and the $\varepsilon_3$ increases first, then decreases and then increases and the peak deviational stress increases.

(2) Under different loading and unloading rates, with the increase in loading and unloading rates, the damage of sandstone specimens is mainly from the online elastic damage end point to the peak point, and the peak damage value decreases and $D_{th}$, $D_c$, $D_b$, $D_a$ increase with the increase in the loading rate. The peak damage values increase and $D_{th}$, $D_c$, $D_b$, $D_a$ decrease with the increase in the unloading rate.

(3) With the increase in the loading rate, the stress brittle drop coefficient of sandstone increases, and the brittle failure weakens. With the increase in the unloading rate, the stress brittle drop coefficient of sandstone decreases first and then increases, and the brittle failure of the rock first becomes stronger first and then weaker. However, the stress brittle drop coefficient appears abnormal at a high unloading rate.

(4) With the increase in the loading rate, $K_{\varepsilon 1}$ first decreases and then increases, and $K_{\varepsilon 2}$ gradually decreases, $K_{\varepsilon 3}$ gradually increases, and $K_{\varepsilon V}$ gradually increases. With the increase in the unloading rate, $K_{\varepsilon 1}$ and $K_{\varepsilon V}$ decrease, and the change in $K_{\varepsilon 2}$ and $K_{\varepsilon 3}$ is not obvious.

**Author Contributions:** Conceptualization, Z.N. and B.Y.; methodology, B.Y.; formal analysis, L.Q.; investigation, X.L.; resources, Y.W.; data curation, D.Z.; writing—original draft preparation, M.W.; writing—review and editing, W.D.; project administration, D.Z.; funding acquisition, D.Z. All authors have read and agreed to the published version of the manuscript.

**Funding:** This research was funded by the National Natural Science Foundation of China (51874053, 52064016), the Scientific Research Foundation of State Key Laboratory of Coal Mine Disaster Dynamics and Control (2011DA105287-zd201804), Jiangxi Provincial Thousand Talents Plan Project (jxsq2019102082).

**Institutional Review Board Statement:** Not applicable.

**Informed Consent Statement:** Informed consent was obtained from all subjects involved in the study.

**Data Availability Statement:** The experimental data supporting the conclusions are available from the corresponding author on request.

**Acknowledgments:** This study was financially supported by the National Natural Science Foundation of China (51874053, 52064016), the Scientific Research Foundation of State Key Laboratory of Coal Mine Disaster Dynamics and Control (2011DA105287-zd201804), Jiangxi Provincial Thousand Talents Plan Project (jxsq2019102082).

**Conflicts of Interest:** The authors declare that they have no known competing financial interests or personal relationships that could have appeared to influence the work reported in this paper.

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
