# Peer review of "The Effects of True Triaxial Loading and Unloading Rates on the Damage Mechanical Properties of Sandstone"

_sustainability, doi:10.3390/su141911899_

Round 1
Reviewer 1 Report
The article contains interesting laboratory tests which, both from a scientific and technological point of view, constitute the basis for the behavior of rocks under high pressure conditions.
Below are some comments and suggestions:
1. In the introduction, a few sentences should be added regarding the numerical modeling of rock failure for different pressures.
2. In the second chapter, the description of the test samples should be extended, in particular: what was the bulk density and whether the samples were cut perpendicularly or parallel to the stratification.
3. The second figure is better called stress-strain characteristics.
4. In subsection 3.3.1 for Figure 10, it should be written what percentage of energy is the post-destructive characteristic. Is it characteristic for sandstone samples?
5. In the discussion section, several references should be added to compare the results of the laboratory tests.
6. In the summary, one conclusion should be added regarding the numerical values ​​presented in the third chapter.
Reviewer 2 Report
Different true triaxial loading and unloading rates are designed in this paper, and the mechanical properties of sandstone under σ1 loading and σ3 unloading are experimentally studied, and some meaningful conclusions are obtained. The paper is clear and concise. This paper meets the publication requirements of this journal, and I recommend a minor revision for it.
Some improvements to be made:
1) The abstract could be more concise.
2) The introduction section should add the latest achievements of international journals, point out the current research situation, and highlight the research content and innovation of this paper.
3) The ViG and ViH formats in lines 174 to 181 should be modified to comply with journal requirements.
4) The conclusion should be concise and highlight the key points.
5) Statement “As Can be seen from Table 6, sandstone at different unloading rate under the condition of true triaxial, as the direction loading and the direction unloading, basically doesn’t produce damage before Dth elastic limit point.” is not clear. It should be modified.
